# Reinforcement Learning with LTL and $\omega$-Regular Objectives via Optimality-Preserving Translation to Average Rewards

**Xuan-Bach Le**[1*]   **Dominik Wagner**[1*]
**Leon Witzman**[1]   **Alexander Rabinovich**[2]   **Luke Ong**[1]

[1]NTU Singapore   [2]Tel Aviv University

{bach.le,dominik.wagner,luke.ong}@ntu.edu.sg
witz0001@e.ntu.edu.sg   rabinoa@tauex.tau.ac.il

## Abstract

Linear temporal logic (LTL) and, more generally, $\omega$-regular objectives are alternatives to the traditional discount sum and average reward objectives in reinforcement learning (RL), offering the advantage of greater comprehensibility and hence explainability. In this work, we study the relationship between these objectives. Our main result is that each RL problem for $\omega$-regular objectives can be reduced to a limit-average reward problem in an optimality-preserving fashion, via (finite-memory) reward machines. Furthermore, we demonstrate the efficacy of this approach by showing that optimal policies for limit-average problems can be found asymptotically by solving a sequence of discount-sum problems approximately. Consequently, we resolve an open problem: optimal policies for LTL and $\omega$-regular objectives can be learned asymptotically.

## 1   Introduction

Reinforcement learning (RL) is a machine learning paradigm whereby an agent aims to accomplish a task in a generally unknown environment [37]. Traditionally, tasks are specified via a scalar reward signal obtained continuously through interactions with the environment. These rewards are aggregated over entire trajectories either through averaging or by summing the exponentially decayed rewards. However, in many applications, there are no reward signals that can naturally be extracted from the environment. Moreover, reward signals that are supplied by the user are prone to error in that the chosen low-level rewards often fail to accurately capture high-level objectives. Generally, policies derived from local rewards-based specifications are hard to understand because it is difficult to express or explain their global intent.

As a remedy, it has been proposed to specify tasks using formulas in Linear Temporal Logic (LTL) [41, 30, 9, 38, 15, 34, 14] or $\omega$-regular languages more generally [30]. In this framework, the aim is to maximise the probability of satisfying a logical specification. LTL can precisely express a wide range of high-level behavioural properties such as liveness (infinitely often $P$), safety (always $P$), stability (eventually always $P$), and priority ($P$ then $Q$ then $T$).

Motivated by this, a growing body of literature study learning algorithms for RL with LTL and $\omega$-regular objectives (e.g. [41, 15, 30, 7, 32, 20, 21, 16]). However, to the best of our knowledge, all of these approaches may fail to learn provably optimal policies without prior knowledge of a generally unknown parameter such as the optimal $\epsilon$-return mixing time [15] or the $\epsilon$-recurrence time [30],

---

*These authors contributed equally to this work.

which depend on the (unavailable) transition probabilities of the MDP. Moreover, it is known that neither LTL nor (limit) average reward objectives are PAC (probably approximately correct) learnable [2]. Consequently, approximately optimal policies can only possibly be found asymptotically but not in bounded time. [1]

In this work, we pursue a different strategy: rather than solving the RL problem directly, we study *optimality-preserving* translations [2] from $\omega$-regular objectives to more traditional rewards, in particular, limit-average rewards. This method offers a significant advantage: it enables the learning of optimal policies for $\omega$-regular objectives by solving a single more standard problem, for which we can leverage existing off-the-shelf algorithms (e.g. [26, 15, 30]). In this way, all future advances—in both theory and practice—for these much more widely studied problems carry over directly, whilst still enjoying significantly more explainable and comprehensible specifications. It is well-known that such a translation from LTL to discounted rewards is impossible [2]. Intuitively, this is because the latter cannot capture infinite horizon tasks such as reachability or safety [2, 42, 19]. Hence, we instead investigate translations to limit-average rewards in this paper.

**Contributions**

We study reinforcement learning of $\omega$-regular and LTL objectives in Markov decision processes (MDPs) with unknown probability transitions, translations to limit-average reward objectives and learning algorithms for the latter. In detail:

1. We prove a negative result (Proposition 4): in general it is not possible to translate $\omega$-regular objectives to limit average objectives in an optimality-preserving manner if rewards are memoryless (i.e., independent of previously performed actions, sometimes called history-free or Markovian).
2. On the other hand, our main result (Theorem 12) resolves Open Problem 1 in [2]: such an optimality-preserving translation is possible if the reward assignment may use finite memory as formalised by reward machines [23, 24].
3. To underpin the efficacy of our reduction approach, we provide the first convergence proof (Theorem 16) of an RL algorithm (Algorithm 1) for average rewards. To the best of our knowledge (and as indicated by [13]), this is the first proof *without assumptions on the induced Markov chains*. In particular, the result applies to multichain MDPs, which our translation generally produces, with unknown probability transitions. Consequently, we also resolve Open Problem 4 of [2]: RL for $\omega$-regular and LTL objectives can be learned in the limit (Theorem 18).

**Outline.** We start by reviewing the problem setup in Section 2. Motivated by the impossibility result for simple reward functions, we define reward machines (Section 3). In Section 4 we build intuition for the proof of our main result in Section 5. Thereafter, we demonstrate that RL with limit-average, $\omega$-regular and LTL objectives can be learned asymptotically (Section 6). Finally, we review related work and conclude in Section 7.

## 2 Background

Recall that a *Markov Decision Process (MDP)* is a tuple $\mathcal{M} = (S, A, s_0, P)$ where $S$ is a finite set of states, $s_0 \in S$ is the initial state, $A$ is the finite set of actions and $P : S \times A \times S \to [0, 1]$ is the probability transition function such that $\sum_{s' \in S} P(s, a, s') = 1$ for every $s \in S$ and $a \in A$. MDPs may be graphically represented; see e.g. Fig. 1a. We let $\mathrm{Runs}_{\mathrm{fi}}(S, A) = S \times (A \times S)^*$ and $\mathrm{Runs}(S, A) = (S \times A)^\omega$ denote the set of finite runs and the set of infinite runs in $\mathcal{M}$ respectively.

A *policy* $\pi : \mathrm{Runs}_{\mathrm{fi}}(S, A) \to \mathcal{D}(A)$ maps finite runs to distributions over actions. We let $\Pi(S, A)$ denote the set of all such policies. A policy $\pi$ is *memoryless* if $\pi(s_0 a_0 \ldots s_n) = \pi(s_0' a_0' \ldots s_m')$ for all finite runs $s_0 a_0 \ldots s_n$ and $s_0' a_0' \ldots s_m'$ such that $s_n = s_m'$. For each MDP $\mathcal{M}$ and policy $\pi$, there is a natural induced probability measure $\mathcal{D}_\pi^{\mathcal{M}}$ on its runs.

The desirability of policies for a given MDP $\mathcal{M}$ can be expressed as a function $\mathcal{J} : \Pi(S, A) \to \mathbb{R}$. Much of the RL literature focuses on discounted-sum $\mathcal{J}_{\mathcal{R}^\gamma}^{\mathcal{M}}$ and limit-average reward objectives $\mathcal{J}_{\mathcal{R}^{\mathrm{avg}}}^{\mathcal{M}}$,

---

[1]Formally, for some $\epsilon, \delta > 0$ it is impossible to learn $\epsilon$-approximately optimal policies with probability $1 - \delta$ in finite time.

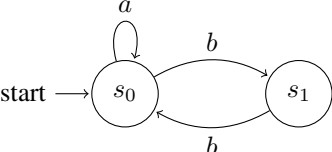

(a) An MDP where all transitions occur with probability 1, $\lambda(s_0, b, s_1) = \{p\}$ and the rest are labeled with $\emptyset$.

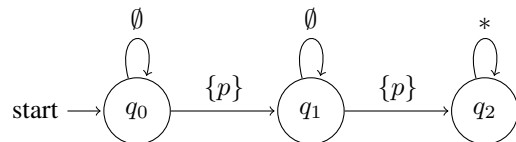

(b) A DRA, where $F := \{(\{q_1\}, \emptyset)\}$, for the objective to visit the petrol station $p$ exactly once.

Figure 1: Examples of an MDP and DRA.

which lift a reward function $\mathcal{R} : S \times A \times S \to \mathbb{R}$ for single transitions to runs $\rho = s_0 a_0 s_1 a_1 \ldots$ as follows:

$$\mathcal{J}^{\mathcal{M}}_{\mathcal{R}^\gamma}(\pi) := \mathbb{E}_{\rho \sim \mathcal{D}^{\mathcal{M}}_\pi} \left[ \sum_{i=0}^{\infty} \gamma^i \cdot r_i \right] \qquad \mathcal{J}^{\mathcal{M}}_{\mathcal{R}^{\text{avg}}}(\pi) := \liminf_{t \to \infty} \mathbb{E}_{\rho \sim \mathcal{D}^{\mathcal{M}}_\pi} \left[ \frac{1}{t} \cdot \sum_{i=0}^{t-1} r_i \right]$$

where $r_i = \mathcal{R}(s_i, a_i, s_{i+1})$ and $\gamma \in (0, 1)$ is the *discount factor*.

**$\omega$-Regular Objectives.** $\omega$-regular objectives (which subsume LTL objectives) are an alternative to these traditional objectives. Henceforth, we fix an alphabet $\mathcal{AP}$ and a *label function* $\lambda : S \times A \times S \to 2^{\mathcal{AP}}$ for transitions, where $2^X$ is the power set of a set $X$. Each run $\rho = s_0 a_0 s_1 a_1 s_2 \ldots$ induces a sequence of labels $\lambda(\rho) = \lambda(s_0, a_0, s_1) \lambda(s_1, a_1, s_2) \ldots$. Thus, for a set $L \subseteq (2^{\mathcal{AP}})^\omega$ of "desirable" label sequences we can consider the probability of a run's labels being in that set: $\mathbb{P}_{\rho \sim \mathcal{D}^{\mathcal{M}}_\pi}[\lambda(\rho) \in L]$.

**Example 1.** For instance, an autonomous car may want to "visit a petrol station exactly once" to conserve resources (e.g. time or petrol). Consider the MDP in Fig. 1a where the state $s_1$ represents a petrol station. We let $\mathcal{AP} = \{p\}$ ($p$ for petrol), $\lambda(s_0, b, s_1) = \{p\}$, and the rest are labeled with $\emptyset$. The desirable label sequences are $L = \{\lambda_1 \lambda_2 \cdots \mid$ for exactly one $i \in \mathbb{N}, \lambda_i = \{p\}\}$.

In this work, we focus on $L$ which are $\omega$-regular languages. It is well known that $\omega$-regular languages are precisely the languages recognised by Deterministic Rabin Automata (DRA) [27, 29]:

**Definition 2.** A DRA is a tuple $\mathcal{A} = (Q, 2^{\mathcal{AP}}, q_0, \delta, F)$ where $Q$ is a finite state set, $2^{\mathcal{AP}}$ is the alphabet, $q_0 \in Q$ is the initial state, $\delta : Q \times 2^{\mathcal{AP}} \to Q$ is the transition function, and $F = \{(A_1, R_1), \ldots, (A_n, R_n)\}$, where $A_i, R_i \subseteq Q$, is the accepting condition. Let $\rho \in (2^{\mathcal{AP}})^\omega$ be an infinite run and $\text{InfS}(\rho)$ the set of states visited infinitely often by $\rho$. We say $\rho$ is accepted by $\mathcal{A}$ if there exists some $(A_i, R_i) \in F$ such that $\rho$ visits some state in $A_i$ infinitely often whilst visiting every state in $R_i$ finitely often, i.e. $\text{InfS}(\rho) \cap A_i \neq \emptyset$ and $\text{InfS}(\rho) \cap R_i = \emptyset$.

For example, the objective in Example 1 may be represented by the DRA in Fig. 1b.

Thus, the desirability of $\pi$ is the probability of $\pi$ generating an accepting sequence in the DRA $\mathcal{A}$:

$$\mathcal{J}^{\mathcal{M}}_{\mathcal{A}}(\pi) := \mathbb{P}_{\rho \sim \mathcal{D}^{\mathcal{M}}_\pi}[\lambda(\rho) \text{ is accepted by the automaton } \mathcal{A}] \tag{1}$$

**Remarks.** The class of $\omega$-regular languages subsumes languages expressed by Linear Temporal Logic (LTL, see e.g. [4, Ch. 5]), a logical framework in which e.g. reachability (eventually $P$, $\Diamond P$), safety (always $P$, $\Box P$) and reach-avoid (eventually $P$ whilst avoiding $Q$, $(\neg Q) \, \mathcal{U} \, P$) properties can be expressed concisely and intuitively. The specification of our running Example 1 to visit the petrol station exactly once can be expressed as the LTL formula $(\neg p) \, \mathcal{U} \, (p \wedge \bigcirc \Box \neg p)$, where $\bigcirc Q$ denotes "$Q$ holds at the next step". Furthermore, our label function $\lambda$, which maps transitions to labels, is more general than other definitions (e.g. [41, 15, 30]) instead mapping states to labels. As a result, we are able to articulate properties that involve actions, such as "to reach the state $s$ while avoiding taking the action $a$".

**Optimality-Preserving Specification Translations.** Rather than solving the problem of synthesising optimal policies for Eq. (1) directly, we are interested in reducing it to more traditional RL

problems and applying off-the-shelf RL algorithms to find optimal policies. To achieve this, the reduction needs to be *optimality preserving*[2]:

**Definition 3** ([2]). An *optimality-preserving specification translation* from $\omega$-regular objectives to limit-average rewards is a computable function mapping each tuple $(S, A, \lambda, \mathcal{A})$ to $\mathcal{R}_{(S,A,\lambda,\mathcal{A})}$ s.t.

$$\text{policies maximising } \mathcal{J}_{\mathcal{R}^{\text{avg}}}^{\mathcal{M}} \text{ also maximise } \mathcal{J}_{\mathcal{A}}^{\mathcal{M}}, \text{ where } \mathcal{R} := \mathcal{R}_{(S,A,\lambda,\mathcal{A})}$$

for every MDP $\mathcal{M} = (S, A, s_0, P)$, label function $\lambda : S \times A \times S \to 2^{\mathcal{AP}}$ and DRA $\mathcal{A}$.

We stress that since the probability transition function $P$ is generally not known, the specification translation may not depend on it.

## 3 Negative Result and Reward Machines

Reward functions emit rewards purely based on the transition being taken without being able to take the past into account. On the other hand, DRAs have finite memory. Therefore, there cannot generally be optimality-preserving translations from $\omega$-regular objectives to limit average rewards provided by reward functions:

**Proposition 4.** *There is an MDP $\mathcal{M}$ and an $\omega$-regular language $L$ for which it is impossible to find a reward function $\mathcal{R} : S \times A \times S \to \mathbb{R}$ such that every $\mathcal{J}_{\mathcal{R}^{\text{avg}}}^{\mathcal{M}}$-optimal policy of $\mathcal{M}$ also maximises the probability of membership in $L$.*

Remarkably, this rules out optimality-preserving specification translations even if transition probabilities are fully known[3].

*Proof.* Consider the deterministic MDP in Fig. 1a and the objective of Example 1 "to visit $s_1$ exactly once" expressed by the DRA $\mathcal{A}$ in Fig. 1b. Assume towards contradiction there exists a reward function $\mathcal{R} : S \times A \times S \to \mathbb{R}$ such that optimal policies w.r.t. $\mathcal{J}_{\mathcal{R}^{\text{avg}}}^{\mathcal{M}}$ maximise acceptance by $\mathcal{A}$. Note that every policy $\pi^*$ maximising acceptance by the DRA induces the run $s_0(as_0)^n b s_1 b s_0(as_0)^{\omega}$ for some $n \in \mathbb{N}$, and $\mathcal{J}_{\mathcal{A}}^{\mathcal{M}}(\pi^*) = 1$. Thus, its limit-average reward is $\mathcal{J}_{\mathcal{R}^{\text{avg}}}^{\mathcal{M}}(\pi^*) = \mathcal{R}(s_0, a, s_0)$. Now, consider the policy $\pi$ always selecting action $a$ with probability 1. As the run induced by $\pi$ is $s_0(as_0)^{\omega}$, we deduce that $\mathcal{J}_{\mathcal{A}}^{\mathcal{M}}(\pi) = 0$ and $\mathcal{J}_{\mathcal{R}^{\text{avg}}}^{\mathcal{M}}(\pi) = \mathcal{R}(s_0, a, s_0) = \mathcal{J}_{\mathcal{R}^{\text{avg}}}^{\mathcal{M}}(\pi^*)$, which is a contradiction since $\pi$ is not $\mathcal{J}_{\mathcal{A}}^{\mathcal{M}}$-optimal. $\square$

Since simple reward functions lack the expressiveness to capture $\omega$-regular objectives, we employ a generalisation, reward machines [23, 24], whereby rewards may also depend on an internal state:

**Definition 5.** A *reward machine (RM)* is a tuple $\mathcal{R} = (U, u_0, \delta_u, \delta_r)$ where $U$ is a finite set of states, $u_0 \in U$ is the initial state, $\delta_r : U \times (S \times A \times S) \to \mathbb{R}$ is the reward function, and $\delta_u : U \times (S \times A \times S) \to U$ is the update function.

Intuitively, a RM $\mathcal{R}$ utilises the current transition to update its states through $\delta_u$ and assigns the rewards through $\delta_r$. For example, Fig. 2a depicts a reward machine for the MDP of Fig. 1a, where the states count the number of visits to $s_1$ (0 times, once, more than once).

Let $\rho = s_0 a_0 s_1 \cdots$ be an infinite run. Since $\delta_u$ is deterministic, it induces a sequence $u_0 u_1 \ldots$ of states in $\mathcal{R}$, where $e_i = (s_i, a_i, s_{i+1})$ and $u_{i+1} = \delta_u(u_i, e_i)$. The *limit-average reward* of a policy $\pi$ is defined as:

$$\mathcal{J}_{\mathcal{R}^{\text{avg}}}^{\mathcal{M}}(\pi) := \liminf_{t \to \infty} \mathbb{E}_{\rho \sim \mathcal{D}_{\pi}^{\mathcal{M}}} \left[ \frac{1}{t} \sum_{i=0}^{t-1} \delta_r(u_i, e_i) \right]$$

It is seen that limit-average optimal policies $\pi^*$ for the MDP in Fig. 1a and the RM in Fig. 2a eventually select action $b$ exactly once in state $s_0$ to achieve $\mathcal{J}_{\mathcal{R}^{\text{avg}}}^{\mathcal{M}}(\pi^*) = 1$.

In the following two sections, we present a general translation from $\omega$-regular languages to limit-average reward machines, and we show that our translation is optimality-preserving (Theorem 12).

---

[2]This definition makes sense both for the case of reward functions $\mathcal{R}_{(S,A,\lambda,\mathcal{A})} : S \times A \times S \to \mathbb{R}$ and reward machines (introduced in the subsequent section).

[3]In Appendix A we show another negative result (Proposition 19): even for a strict subset of $\omega$-regular specifications such translations are impossible.

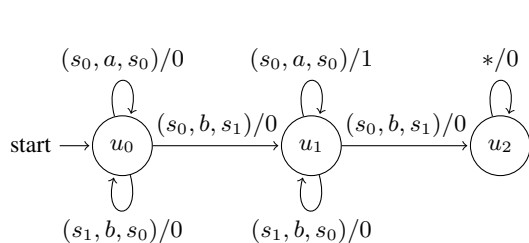

(a) A reward machine for the objective of visiting the petrol station exactly once. (The rewards are given following "/".)

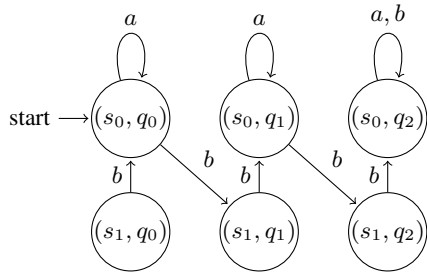

(b) Product MDP for Fig. 1, where all transitions have probability 1 and $F_\mathcal{M} := \{(\{(s_0, q_1), (s_1, q_1)\}, \emptyset)\}$.

Figure 2: A reward machine and the product MDP for the running Example 1.

**Remarks.** Our definition of RM is more general than the one presented in [23, 24], where $\delta'_u : U \to [S \times A \times S \to \mathbb{R}]$ and $\delta'_r : U \times 2^{\mathcal{AP}} \to U$. Note that $(\delta'_u, \delta'_r)$ can be reduced to $(\delta_u, \delta_r)$ by expanding the state space of the RM to include the previous state and utilising the inverse label function $\lambda^{-1}$. It is worth pointing out that Theorem 12 does not contradict a negative result in [2] regarding the non-existence of an optimality-preserving translation from LTL constraints to *abstract* limit-average reward machines (where only the *label* of transitions is provided to $\delta_u$ and $\delta_r$).

## 4 Warm-Up: Transitions with Positive Probability are Known

To help the reader gain intuition about our construction, we first explore the situation where the support $\{(s, a, s') \in S \times A \times S \mid P(s, a, s') > 0\}$ of the MDP's transition function is known. Crucially, we do not assume that the *magnitude* of these (non-zero) probabilities are known. Subsequently, in Section 5, we fully eliminate this assumption.

This assumption allows us to draw connections between our problem and a familiar scenario in probabilistic model checking [4, Ch. 10], where the acceptance problem for $\omega$-regular objectives can be transformed into a reachability problem. Intuitively, our reward machine monitors the state of the DRA and provides reward 1 if the MDP and the DRA are in certain "good" states (0 otherwise).

For the rest of this section, we fix an MDP without transition function $(S, A, s_0)$, a set of possible transitions $E \subseteq S \times A \times S$, a label function $\lambda : S \times A \times S \to 2^{\mathcal{AP}}$ and a DRA $\mathcal{A} = (Q, 2^{\mathcal{AP}}, q_0, \delta, F)$. Our aim is to find a reward machine $\mathcal{R}$ such that for every transition function $P$ compatible with $E$ (formally: $E = \{(s, a, s') \mid P(s, a, s') > 0\}$), optimal policies for limit-average rewards are also optimal for the acceptance probability of the DRA $\mathcal{A}$.

### 4.1 Product MDP and End Components

First, we form the *product MDP* $\mathcal{M} \otimes \mathcal{A}$ (e.g. [41, 15]), which synchronises the dynamics of the MDP $\mathcal{M}$ with the DRA $\mathcal{A}$. Formally, $\mathcal{M} \otimes \mathcal{A} = (V, A, v_0, \Delta, F_\mathcal{M})$ where $V = S \times Q$ is the set of states, $A$ is the set of actions, $v_0 = (s_0, q_0)$ is the initial state. The transition probability function $\Delta : V \times A \times V \to [0, 1]$ satisfies $\Delta(v, a, v') = P(s, a, s')$ given that $v = (s, q)$, $v' = (s', q')$, and $\delta(q, \lambda(s, a, s')) = q'$. The accepting condition is $F_\mathcal{M} = \{(A'_1, R'_1), (A'_2, R'_2), \ldots\}$ where $A'_i = S \times A_i$, $R'_i = S \times R_i$, and $(A_i, R_i) \in F$. A run $\rho = (s_0, q_0) a_0 \cdots$ is accepted by $\mathcal{M} \otimes \mathcal{A}$ if there exists some $(A'_i, R'_i) \in F_\mathcal{M}$ such that $\text{InfV}(\rho) \cap A'_i \neq \emptyset$ and $\text{InfV}(\rho) \cap R'_i = \emptyset$, where $\text{InfV}$ is the set of states $(s, v)$ in the product MDP visited infinitely often by $\rho$.

Note that product MDPs have characteristics of both MDPs and DRAs which neither possesses in isolation: transitions are generally probabilistic and there is a notation of acceptance of runs. For example, the product MDP for Fig. 1 is shown in Fig. 2b. Due to the deterministic nature of the DRA $\mathcal{A}$, every run $\rho$ in $\mathcal{M}$ gives rise to a unique run $\rho^\otimes$ in $\mathcal{M} \otimes \mathcal{A}$. Crucially, for every policy $\pi$,

$$\mathbb{P}_{\rho \sim \mathcal{D}^\mathcal{M}_\pi}[\rho \text{ is accepted by } \mathcal{A}] = \mathbb{P}_{\rho \sim \mathcal{D}^\mathcal{M}_\pi}[\rho^\otimes \text{ is accepted by } \mathcal{M} \otimes \mathcal{A}] \tag{2}$$

We make use of well-known almost-sure characterisation of accepting runs via the notion of accepting end components:

**Definition 6.** An *end component* (EC) of $\mathcal{M} \otimes \mathcal{A} = (V, A, v_0, \Delta, F_{\mathcal{M}})$ is a pair $(T, \mathrm{Act})$ where $T \subseteq V$ and $\mathrm{Act} : T \to 2^A$ satisfies the following conditions

1. For every $v \in T$ and $a \in \mathrm{Act}(v)$, we have $\sum_{v' \in T} \Delta(v, a, v') = 1$, and
2. The graph $(T, \to_{\mathrm{Act}})$ is strongly connected, where $v \to_{\mathrm{Act}} v'$ iff $\Delta(v, a, v') > 0$ for some $a \in \mathrm{Act}(v)$.

$(T, \mathrm{Act})$ is an *accepting EC (AEC)* if $T \cap A_i' \neq \emptyset$ and $T \cap R_i' = \emptyset$ for some $(A_i', R_i') \in F_{\mathcal{M}}$.

Intuitively, an EC is a strongly connected sub-MDP. For instance, for the product MDP in Fig. 2b there are five end components, $(\{(s_0, q_0)\}, (s_0, q_0) \mapsto \{a\})$, $(\{(s_0, q_1)\}, (s_0, q_1) \mapsto \{a\})$, $(\{(s_0, q_2)\}, (s_0, q_2) \mapsto \{a\})$, $(\{(s_0, q_2)\}, (s_0, q_2) \mapsto \{b\})$ and $(\{(s_0, q_2)\}, (s_0, q_2) \mapsto \{a, b\})$. $(\{(s_0, q_1)\}, (s_0, q_1) \mapsto \{a\})$ is its only accepting end component.

It turns out that, almost surely, a run is accepted iff it enters an accepting end component and never leaves it [1]. Therefore, a natural idea for a reward machine is to use its state to keep track of the state $q \in Q$ the DRA is in and give reward 1 to transitions $(s, a, s')$ if $(s, q)$ is in some AEC (and 0 otherwise). Unfortunately, this approach falls short since the AEC may contain non-accepting ECs, thus assigning maximal reward to sub-optimal policies.[4] As a remedy, we introduce a notion of minimal AEC, and ensure that only runs eventually committing to one such minimal AEC get a limit-average reward of 1.

**Definition 7.** An AEC $(T, \mathrm{Act})$ is an *accepting simple EC (ASEC)* if $|\mathrm{Act}(v)| = 1$ for every $v \in T$.

Let $\mathcal{C}_1 = (T_1, \mathrm{Act}_1), \ldots, \mathcal{C}_n = (T_n, \mathrm{Act}_n)$ be a collection of ASECs covering all states in ASECs, i.e. if $(s, q)$ is in some ASEC then $(s, q) \in T_1 \cup \cdots \cup T_n$. In particular, $n \leq |S \times Q|$ is sufficient.

We can prove that every AEC contains an ASEC (see Lemma 20 in Appendix B). Consequently,

**Lemma 8.** *Almost surely, if $\rho$ is accepted by $\mathcal{A}$ then $\rho^{\otimes}$ reaches a state in some ASEC $\mathcal{C}_i$ of $\mathcal{M} \otimes \mathcal{A}$.*

### 4.2 Reward Machine and Correctness

Next, to ensure that runs eventually commit to one such ASEC we introduce the following notational shorthand: for $(s, q) \in T_1 \cup \cdots \cup T_n$, let $\mathcal{C}_{(s,q)} = (T_{(s,q)}, \mathrm{Act}_{(s,q)})$ be the $\mathcal{C}_i$ with minimal $i$ containing $(s, q)$, i.e. $C_{(s,q)} := C_{\min\{1 \leq i \leq n | (s,q) \in T_i\}}$.

Intuitively, we give a reward of 1 if $(s, q)$ is in one of the $\mathcal{C}_1, \ldots, \mathcal{C}_n$. However, once an action is performed which deviates from $\mathrm{Act}_{(s,q)}$ no rewards are given thereafter, thus resulting in a limit average reward of 0.

A state in the reward machine has the form $q \in Q$, keeping track of the state in the DRA, or $\perp$, which is a sink state signifying that in a state in $\mathcal{C}_1, \ldots, \mathcal{C}_n$ we have previously deviated from $\mathrm{Act}_{(s,q)}$.

Finally, we are ready to formally define the reward machine $\mathcal{R} = \mathcal{R}_{(S, A, \lambda, \mathcal{A})}$ exhibiting our specification translation as $(Q \cup \{\perp\}, q_0, \delta_u, \delta_r)$, where

$$\delta_u(u, (s, a, s')) := \begin{cases} \perp & \text{if } u = \perp \text{ or} \\ & ((s, u) \in T_1 \cup \cdots \cup T_n \text{ and } a \notin \mathrm{Act}_{(s,u)}(s, u)) \\ \delta(u, \lambda(s, a, s')) & \text{otherwise} \end{cases}$$

$$\delta_r(u, (s, a, s')) := \begin{cases} 1 & \text{if } u \neq \perp \text{ and } (s, u) \in T_1 \cup \cdots \cup T_n \\ 0 & \text{otherwise} \end{cases}$$

For our running example, this construction essentially yields the reward machine in Fig. 2a (with some inconsequential modifications cf. Fig. 4 in Appendix B).

**Theorem 9.** *For all transition probability functions $P$ with support $E$, policies maximising the limit-average reward w.r.t. $\mathcal{R}$ also maximise the acceptance probability of the DRA $\mathcal{A}$.*

This result follows immediately from the following (the full proof is presented in Appendix B):

---

[4]To illustrate this point, consider the product MDP $(\{s_0, s_1\}, \{a, b\}, s_0, P, F)$ where $P(s_0, b, s_0) = P(s_0, a, s_1) = P(s_1, a, s_0) = 1$ and $F = \{(\{s_1\}, \emptyset)\}$, i.e. the objective is to visit $s_1$ infinitely often.

**Lemma 10.** *Let $P$ be a probability transition function with support $E$ and $\mathcal{M} := (S, A, s_0, P)$.*

    *1. For every policy $\pi$, $\mathcal{J}^{\mathcal{M}}_{\mathcal{R}^{avg}}(\pi) \leq \mathcal{J}^{\mathcal{M}}_{\mathcal{A}}(\pi)$.*

    *2. For every policy $\pi$, there exists some policy $\pi'$ satisfying $\mathcal{J}^{\mathcal{M}}_{\mathcal{A}}(\pi) \leq \mathcal{J}^{\mathcal{M}}_{\mathcal{R}^{avg}}(\pi')$.*

*Proof sketch.* 1. By construction, every run receiving a limit-average reward of 1, must have entered some ASEC $\mathcal{C}_i$ and never left it. Furthermore, almost surely all states are visited infinitely often and the run is accepted by definition of accepting ECs.

2. By Lemma 8, almost surely, a run is only accepted if it enters some $\mathcal{C}_i$. We set $\pi'$ to be the policy agreeing with $\pi$ until reaching one of the $\mathcal{C}_1, \ldots, \mathcal{C}_n$ and henceforth following the action $\mathrm{Act}_{(s_t, q_t)}(s_t, q_t)$, where $q_t$ is the state of the DRA at step $t$, yielding a guaranteed limit-average reward of 1 for the run by construction. $\qquad\square$

**Remark 11.** Our construction considers a collection of ASECs covering all states in ASECs. Whilst it does not necessarily require listing all possible ASECs but only (up to) one ASEC per state, it is unclear whether this can be obtained in polynomial time. In Appendix B.1, we present an alternative (yet more complicated) construction which has polynomial time complexity.

## 5 Main Result

In this section, we generalise the approach of the preceding section to prove our main result:

**Theorem 12.** *There exists an optimality-preserving translation from $\omega$-regular languages to limit-average reward machines.*

Again, we fix an MDP without transition function $(S, A, s_0)$, a label function $\lambda : S \times A \times S \to 2^{\mathcal{AP}}$ and a DRA $\mathcal{A} = (Q, 2^{\mathcal{AP}}, q_0, \delta, F)$. Note that the ASECs of a product MDP are uniquely determined by the non-zero probability transitions. Thus, for each set of transitions $E \subseteq (S \times Q) \times A \times (S \times Q)$, we let $\mathcal{C}^E_1 = (T_1, \mathrm{Act}_1), \ldots, \mathcal{C}^E_n = (T_n, \mathrm{Act}_n)$ denote a collection of ASECs covering all states in ASECs w.r.t. the MDPs in which $E$ is the set of non-zero probability transitions.[5] Then, for each set $E$ and state $(s, q) \in T^E_1 \cup \cdots \cup T^E_n$, we let $\mathcal{C}^E_{(s,q)} = (T^E_{(s,q)}, \mathrm{Act}^E_{(s,q)})$ be the ASEC $\mathcal{C}^E_i$ that contains $(s, q)$ in which the index $i$ is minimal.

Our reward machine $\mathcal{R} = \mathcal{R}_{(S,A,\lambda,\mathcal{A})}$ extends the ideas from the preceding section. Importantly, we keep track of the set of transitions $E$ taken so far and assign rewards according to our current knowledge about the graph of the product MDP. Therefore, we propose employing states of the form $(q, f, E)$, where $q \in Q$ keeps track of the state of the DRA, $f \in \{\top, \bot\}$ is a *status flag* and $E \subseteq (S \times Q) \times A \times (S \times Q)$ memorises the transitions in the product MDP encountered thus far.

Intuitively, we set the flag to $\bot$ if we are in MDP state $s$, $(s, q)$ is in one of the $\mathcal{C}^E_1, \ldots, \mathcal{C}^E_n$ and the chosen action deviates from $\mathrm{Act}^E_{(s,q)}(s, q)$. We can recover from $\bot$ by discovering new transitions. Besides, we give reward 1 if $f = \top$ and $(s, q)$ is in one of the $\mathcal{C}^E_1, \ldots, \mathcal{C}^E_n$ (and 0 otherwise).

The status flag is required since discovering new transitions will change the structure of (accepting simple) end components. Hence, differently from the preceding section, it is not sufficient to have a single sink state.

The initial state of our reward machine is $u_0 := (q_0, \top, \emptyset)$ and we formally define the update and reward functions as follows:

$$\delta_u((q, f, E), (s, a, s')) := \begin{cases} (q', \bot, E) & \text{if } f = \bot \text{ and } e \in E \\ (q', \bot, E) & \text{if } f = \top, e \in E, (s, q) \in T^E_1 \cup \cdots \cup T^E_n \text{ and} \\ & a \notin \mathrm{Act}^E_{(s,q)}(s, q) \\ (q', \top, E \cup \{e\}) & \text{otherwise} \end{cases}$$

$$\delta_r((q, f, E), (s, a, s')) := \begin{cases} 1 & \text{if } f = \top, (s, q) \in T^E_1 \cup \cdots \cup T^E_n \\ 0 & \text{otherwise} \end{cases}$$

where $q' := \delta(q, \lambda(s, a, s'))$ and $e := ((q, s), a, (q', s'))$.

---

[5]To achieve the same number $n$ of ASECs we can add duplicates. If there are no ASECs we can set $T_i := \emptyset$.

**Example 13.** For our running example (see Example 1 and Fig. 1) initially no transitions are known (hence no ASECs). Therefore, all transitions receive reward 0. Once action $a$ has been performed in state $s_0$ in the MDP $\mathcal{M}$ and $(q_1, f, E)$ in the reward machine $\mathcal{R}$, we have discovered the ASEC $(\{(s_0, q_1)\}, (s_0, q_1) \mapsto \{a\})$ and a reward of 1 is given henceforth unless action $b$ is selected eventually. In that case, we leave the ASEC and we will not discover further ASECs since there is only one. From here, it is not possible to return to state $q_1$ in the DRA and henceforth only reward 0 will be obtained.

Theorem 12 is proven by demonstrating an extension of Lemma 10 (see Appendix C):

**Lemma 14.** *Suppose $\mathcal{M} = (S, A, s_0, P)$ is an arbitrary MDP.*

1. *For every policy $\pi$, $\mathcal{J}_{\mathcal{R}^{avg}}^{\mathcal{M}}(\pi) \leq \mathcal{J}_{\mathcal{A}}^{\mathcal{M}}(\pi)$.*
2. *For every policy $\pi$, there exists some policy $\pi'$ satisfying $\mathcal{J}_{\mathcal{A}}^{\mathcal{M}}(\pi) \leq \mathcal{J}_{\mathcal{R}^{avg}}^{\mathcal{M}}(\pi')$.*

Note that Lemma 14 immediately proves that the reduction is not only optimality preserving (Theorem 12) but also robust: every $\epsilon$-approximately limit-average optimal policy is also $\epsilon$-approximately optimal w.r.t. $\mathcal{J}_{\mathcal{A}}^{\mathcal{M}}$. This observation is important because *exactly* optimal policies for the limit average problem may be hard to find.

Intuitively, to see part 1 of Lemma 14 we note: If an average reward of 1 is obtained for a run, the reward machine believes, based on the partial observation of the product MDP, that the run ends up in an ASEC. Almost surely, we eventually discover all possible transitions involving the same state-action pairs as this ASEC and therefore this must also be an ASEC w.r.t. the true, unknown product MDP. For part 2, we modify the policy $\pi$ similarly as in Lemma 10 by selecting actions $\mathrm{Act}(s_t, q_t)$ once having entered an ASEC $\mathcal{C} = (T, \mathrm{Act})$ w.r.t. the true, unknown product MDP.[6]

## 6 Convergence for Limit Average, $\omega$-Regular and LTL Objectives

Thanks to the described translation, advances (in both theory and practice) in the study of RL with average rewards carry over to RL with $\omega$-regular and LTL objectives. In this section, we show that it is possible to learn optimal policies for limit average rewards in the limit. Hence, we resolve an open problem [2]: also RL with $\omega$-regular and LTL objectives can also be learned in the limit.

We start with the case of simple reward functions $\mathcal{R} : S \times A \times S \to \mathbb{R}$. Recently, [18, Theorem 4.2] have shown that discount optimal policies for sufficiently high discount factor $\overline{\gamma} \in [0, 1)$ are also limit average optimal.[7] This result alone is not enough to demonstrate Theorem 16 since $\overline{\gamma}$ is generally not known and in finite time we might only obtain *approximately* limit average optimal policies.

Our approach is to reduce RL with average rewards to a *sequence* of discount sum problems with increasingly high discount factor, which are solved with increasingly high accuracy. Our crucial insight is that eventually the approximately optimal solutions to the discounted problems will also be limit average optimal (see Appendix D for a proof):

**Lemma 15.** *Suppose $\gamma_k \nearrow 1$, $\epsilon_k \searrow 0$ and suppose each $\pi_k$ is a memoryless policy. Then there exists $k_0$ such that for all $K \ni k \geq k_0$, $\pi_k$ is limit average optimal, where $K$ is the set of $k \in \mathbb{N}$ satisfying $\mathcal{J}_{\mathcal{R}^{\gamma_k}}^{\mathcal{M}}(\pi_k) \geq \mathcal{J}_{\mathcal{R}^{\gamma_k}}^{\mathcal{M}}(\pi) - \epsilon_k$ for all memoryless policies $\pi$.*

Our proof harnesses yet another notion of optimality: a policy $\pi$ is *Blackwell optimal* (cf. [6] and [22, Sec. 8.1]) if there exists $\overline{\gamma} \in (0, 1)$ such that $\pi$ is $\gamma$-discount optimal for all $\overline{\gamma} \leq \gamma < 1$. It is well-known that memoryless Blackwell optimal policies always exist [6, 18] and they are also limit-average optimal [22, 18].

Thanks to the PAC (probably approximately correct) learnability of RL with discounted rewards [26, 36], there exists an algorithm `Discounted` which receives as inputs a simulator for $\mathcal{M}, \mathcal{R}$ as well as $\gamma, \epsilon$ and $\delta$, and with probability $1 - \delta$ returns an $\epsilon$-optimal memoryless policy for discount factor $\gamma$. In view of Lemma 15, our approach is to run the PAC algorithm for discount-sum RL for increasingly large discount factors $\gamma$ and increasingly low $\delta$ and $\epsilon$ (Algorithm 1).

---

[6]NB The modified policy depends on the true, unknown support of the probability transition function; we only claim the *existence* of such a policy.

[7]Recall (see e.g. [22, Sec. 8.1]) that for any policy $\pi \in \Pi$, $\lim_{\gamma \nearrow 1}(1 - \gamma) \cdot \mathcal{J}_{\mathcal{R}^\gamma}^{\mathcal{M}}(\pi) = \mathcal{J}_{\mathcal{R}^{avg}}^{\mathcal{M}}(\pi)$.

**Algorithm 1** RL for limit average rewards

---

**Require:** simulator for $\mathcal{M}, \mathcal{R}$
   **for** $k \in \mathbb{N}$ **do**
      $\pi_k \leftarrow \texttt{Discounted}(\mathcal{M}, \mathcal{R}, \underbrace{1 - 1/k}_{\gamma_k}, \underbrace{1/k}_{\epsilon_k}, \underbrace{1/k^2}_{\delta_k})$
   **end for**

---

**Theorem 16.** *RL with average reward functions can be learned in the limit by Algorithm 1: almost surely there exists $k_0 \in \mathbb{N}$ such that $\pi_k$ is limit-average optimal for $k \geq k_0$.*

*Proof.* Using the definition for $K$ of Lemma 15 of iterations where the PAC-MDP algorithm succeeds,

$$\mathbb{E}\left[\#(\mathbb{N} \setminus K)\right] \leq \sum_{k \in \mathbb{N}} \mathbb{P}[\text{PAC-MDP fails in iteration } k] \leq \sum_{k \in \mathbb{N}} \delta_k = \sum_{k \in \mathbb{N}} \frac{1}{k^2} < \infty$$

The claim follows immediately with Lemma 15. $\qquad\square$

Next, we turn to the more general case of reward *machines*. [23, 24] observe that optimal policies for reward machines can be learned by learning optimal policies for the modified MDP which additionally tracks the state the reward machine is in and assigns rewards accordingly. We conclude at once:

**Corollary 17.** *RL with average reward machines can be learned in the limit.*

Finally, harnessing Theorem 12 we resolve Open Problem 4 of [2]:

**Theorem 18.** *RL with $\omega$-regular and LTL objectives can be learned in the limit.*

**Discussion.** Algorithm 1 makes independent calls to black box algorithms for discount sum rewards. Many such algorithms with PAC guarantees are model based (e.g. [26, 36]) and sample from the MDP to obtain suitable approximations of the transition probabilities. Thus, Algorithm 1 can be improved in practice by re-using approximations obtained in earlier iterations and refining them.

## 7  Related Work and Conclusion

The connection between acceptance of $\omega$-regular languages in the product MDP and AECs is well-known in the field of probabilistic model checking [4, 12]. As an alternative to DRAs [41, 14, 32], Limit Deterministic Büchi Automata [35] have been employed to express $\omega$-regular languages for RL [38, 7, 10, 20, 21].

A pioneering work on RL for $\omega$-regular rewards is [41], which expresses $\omega$-regular objectives using Deterministic Rabin Automata. Similar RL approaches for $\omega$-regular objectives can also be found in [14, 38, 10, 15]. The authors of [15, 30] approach RL for $\omega$-regular objectives directly by studying the reachability of AECs in the product MDP and developing variants of the R-MAX algorithm [8] to find optimal policies. However, these approaches require prior knowledge of the MDP, such as the structure of the MDP, the optimal $\epsilon$-return mixing time [15], or the $\epsilon$-recurrence time [30].

Various studies have explored reductions of $\omega$-regular objectives to discounted rewards, and subsequently applied Q-learning and its variants for learning optimal policies [7, 32, 20, 21, 16]. In a similar spirit, [39] present a translation from LTL objectives to *eventual discounted* rewards, where only strictly positive rewards are discounted. These translations are generally not optimality preserving unless the discount factor is selected in a suitable way. Again, this is impossible without prior knowledge of the exact probability transition functions in the MDP.

[25] propose a translation to limit-average rewards for $\omega$-regular specifications which are also *absolute liveness* properties. (In particular, optimal policies satisfy such specifications with either probability 0 or 1.) Their translation is optimality-preserving provided the MDP is *communicating* and the magnitude of penalty rewards in their construction are chosen sufficiently large (which requires knowledge of the MDP).

Furthermore, whilst there are numerous convergent RL algorithms for average rewards for *unichain* or *communicating*[8] MDPs (e.g. [8, 43, 17, 33, 3, 40]), it is unknown whether such an algorithm exists for general multichain MDPs with a guaranteed convergence property. In fact, a negative result in [2, 5] shows that there is no PAC (probably approximately correct) algorithm for LTL objectives and limit-average rewards when the MDP transition probabilities are unknown.

[8] have proposed an algorithm with PAC guarantees provided $\epsilon$-return mixing times are known. They informally argue that for fixed sub-optimality tolerance $\epsilon$, this assumption can be lifted by guessing increasingly large candidates for the $\epsilon$-return mixing time. This yields $\epsilon$-approximately optimal policies in the limit. However, it is not clear how to asymptotically obtain exactly optimal policies as this would require simultaneously decreasing $\epsilon$ and increasing guesses for the $\epsilon$-return mixing time (which depends on $\epsilon$).

**Conclusion.** We have presented an optimality-preserving translation from $\omega$-regular objectives to limit-average rewards furnished by reward machines. As a consequence, off-the-shelf RL algorithms for average rewards can be employed in conjunction with our translation to learn policies for $\omega$-regular objectives. Furthermore, we have developed an algorithm asymptotically learning provably optimal policies for limit-average rewards. Hence, also optimal policies for $\omega$-regular and LTL objectives can be learned in the limit. Our results provide affirmative answers to two open problems in [2].

**Limitations.** We focus on MDPs with finite state and action sets and assume states are fully observable. The assumption of Section 4 that the support of the MDP's probability transition function is known is eliminated in Section 5. Whilst the size of our general translation—the first optimality-preserving translation—is exponential, the additional knowledge in Section 4 enables a construction of the reward machine of the same size as the DRA expressing the objective. Hence, we conjecture that this size is minimal relative to the DRA specification. Since RL with average rewards is not PAC learnable, we cannot possibly provide finite-time complexity guarantees of our Algorithm 1.

## Acknowledgments and Disclosure of Funding

This research is supported by the National Research Foundation, Singapore, under its RSS Scheme (NRFRSS2022-009).

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

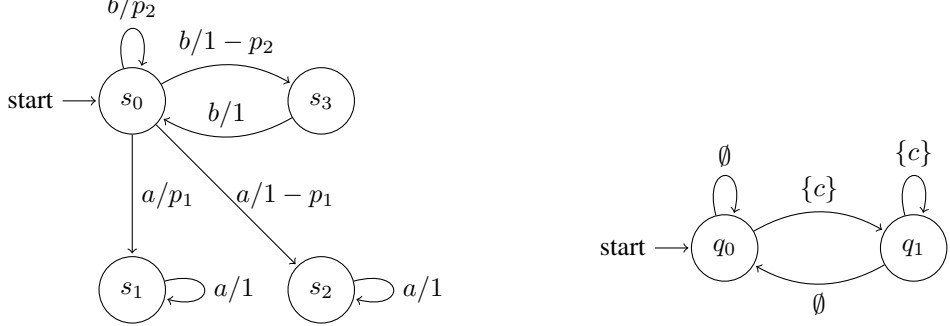

(a) An MDP $\mathcal{M}$ where $\lambda(s_1, a, s_1) = \lambda(s_3, b, s_0) = \{c\}$, and the rest are labeled with $\emptyset$.

(b) A DRA $\mathcal{A}$ for the objective of visiting $s_1$ or $s_3$ infinitely often where $F := \{(\{q_1\}, \emptyset)\}$.

Figure 3: Counter-example for prefix-independent objectives.

## A Supplementary Materials for Section 3

Recall that a $\omega$-regular language $L$ is prefix-independent if for every infinite label sequence $w \in (2^{\mathcal{AP}})^\omega$, we have $w \in L$ iff $w' \in L$ for every suffix $w'$ of $w$. We prove that there is no optimality-preserving translation for reward functions regardless of whether $L$ is prefix-independent or not. The prefix-dependent case was given in Section 3. Here we focus on the other case:

**Proposition 19.** *There exists a tuple $(S, A, s_0, \lambda)$ and a prefix-independent $\omega$-regular language $L$ for which it is impossible to find a reward function $\mathcal{R} : S \times A \times S \to \mathbb{R}$ such that for every probability transition $P$, let $\mathcal{M} = (S, A, s_0, P, \lambda)$, then every $\mathcal{R}^{avg}$-optimal policy of $\mathcal{M}$ is also $L$-optimal (i.e. maximizing the probability of membership in $L$).*

*Proof.* Our proof technique is based on the fact that we can modify the transition probability function. Consider the MDP in Fig. 3a, where the objective is to visit either $s_1$ or $s_3$ infinitely often. It can be checked that the DRA in Fig. 3b captures the given objective and the language accepted by $\mathcal{A}$ is prefix-independent. There are only two deterministic memoryless policies: $\pi_1$, which consistently selects action $a$, and $\pi_2$, which consistently selects action $b$. For the sake of contradiction, let's assume the existence of a reward function $\mathcal{R}$ that preserves optimality for every transition probability function $P$. Pick $p_1 = 1$ and $p_2 = 0$. Then $\mathcal{J}_{\mathcal{A}}^{\mathcal{M}}(\pi_1) = 1$ and $\mathcal{J}_{\mathcal{A}}^{\mathcal{M}}(\pi_2) = 0$, which implies that $\pi_1$ is $\mathcal{A}$-optimal whereas $\pi_2$ is not. Thus $\mathcal{R}(s_1, a, s_1) = \mathcal{J}_{\mathcal{R}^{avg}}^{\mathcal{M}}(\pi_1) > \mathcal{J}_{\mathcal{R}^{avg}}^{\mathcal{M}}(\pi_2) = \mathcal{R}(s_0, b, s_0)$. Now, assume $p_1, p_2 \in (0, 1)$. Accordingly, we have $\mathcal{J}_{\mathcal{R}^{avg}}^{\mathcal{M}}(\pi_1) \geq p_1 \mathcal{R}(s_1, a, s_1)$ and we can deduce that (e.g. by solving the linear equation system described in [31, §8.2.3]) $\mathcal{J}_{\mathcal{R}^{avg}}^{\mathcal{M}}(\pi_2) = \frac{p_2}{2-p_2} \mathcal{R}(s_0, b, s_0) + \frac{1-p_2}{2-p_2} (\mathcal{R}(s_0, b, s_3) + \mathcal{R}(s_3, b, s_0))$. As a result:

$$\lim_{p_1 \to 1} \mathcal{J}_{\mathcal{R}^{avg}}^{\mathcal{M}}(\pi_1) \geq \mathcal{R}(s_1, a, s_1) > \mathcal{R}(s_0, b, s_0) = \lim_{p_2 \to 1} \mathcal{J}_{\mathcal{R}^{avg}}^{\mathcal{M}}(\pi_2)$$

Consequently, if $p_1, p_2$ are sufficiently large then $\mathcal{J}_{\mathcal{R}^{avg}}^{\mathcal{M}}(\pi_1) > \mathcal{J}_{\mathcal{R}^{avg}}^{\mathcal{M}}(\pi_2)$. However, this contradicts to the fact that $\pi_2$ is $\mathcal{A}$-optimal and $\pi_1$ is not, since $\mathcal{J}_{\mathcal{A}}^{\mathcal{M}}(\pi_2) = 1 > p_1 = \mathcal{J}_{\mathcal{A}}^{\mathcal{M}}(\pi_1)$. Hence, there is no such reward function $\mathcal{R}$. $\square$

## B Supplementary Materials for Section 4

**Lemma 20.** *Every AEC contains an ASEC.*

*Proof.* Consider an AEC $\mathcal{C} = (T, \mathrm{Act})$ of $\mathcal{M}_{\mathcal{A}}$. We will prove this by using induction on the number of actions in $\mathcal{C}$, denoted as $\mathsf{size}(\mathcal{C}) := \sum_{s \in T} |\mathrm{Act}(s)| \geq 1$. For the base case where $\mathsf{size}(\mathcal{C}) = 1$, it can be deduced that $\mathcal{C}$ consists of only one accepting state with a self-loop. Therefore, $\mathcal{C}$ itself is an ASEC.

Now, let's assume that $\text{size}(\mathcal{C}) = k + 1 \geq 2$. If $\mathcal{C}$ is already an ASEC, then we are done. Otherwise, there exists a state $s \in T$ such that $|\text{Act}(s)| > 1$. Since $\mathcal{C}$ is strongly connected, there exists a finite path $\rho = sas_1a_1 \ldots s_na_ns_F$ where $s_F$ is an accepting state and all the states $s_1, \ldots, s_n$ are different from $s$. Let $a' \in \text{Act}(s)$ such that $a' \neq a$. We construct a new AEC $\mathcal{C}' = (T', \text{Act}')$ by first removing $a'$ from $\text{Act}(s)$ and then removing all the states that are no longer reachable from $s$ along with their associated transitions. It is important to note that after the removal, $s_F \in T'$ since we can reach $s_F$ from $s$ without taking the action $a'$. (Besides, the graph is still strongly connected.) Since $\text{size}(\mathcal{C}') \leq k$, we can apply the induction hypothesis to conclude that $\mathcal{C}'$ contains an ASEC, thus completing the proof. □

**Lemma 8.** *Almost surely, if $\rho$ is accepted by $\mathcal{A}$ then $\rho^{\otimes}$ reaches a state in some ASEC $\mathcal{C}_i$ of $\mathcal{M} \otimes \mathcal{A}$.*

To proof this result, we recall a well-known result in probabilistic model checking that with probability of one (wpo), every run $\rho$ of the policy $\pi$ eventually stays in one of the ECs of $\mathcal{M}_{\mathcal{A}}$ and visits every transition in that EC infinitely often. To state this formally, we define for any run $\rho = s_0a_0s_1\cdots$,

$$\text{InfSA}(\rho) := \{(s, a) \in S \times A \mid |\{i \in \mathbb{N} \mid s_i = s \wedge a_i = a\}| = \infty\}$$

the set of state-action-pairs occurring infinitely often in $\rho$. Furthermore, a state-action set $\chi \subseteq S \times A$ defines a sub-MDP $\text{sub}(\chi) := (T, \text{Act})$, where

$$T := \{s \in S \mid (s, a) \in \chi \text{ for some } a \in A\} \qquad \text{Act}(s) := \{a \mid (s, a) \in \chi\}$$

**Lemma 21** ([12]). $\mathbb{P}_{\rho \sim \mathcal{D}_\pi^{\mathcal{M} \otimes \mathcal{A}}}[\text{sub}(\text{InfSA}(\rho)) \text{ is an end component}] = 1$.

For the sake of self-containedness, we recall the proof of [12].

*Proof.* We start with two more definitions: for any sub-MDP $(T, \text{Act})$ [1], let

$$\text{sa}(T, \text{Act}) := \{(s, a) \in T \times A \mid a \in \text{Act}(s)\}$$

be the set of state-action pairs $(s, a)$ such that $a$ is enabled in $s$. Finally, let

$$\Omega^{(T, \text{Act})} := \{\rho \in \text{Runs}(S, A) \mid \text{InfSA}(\rho) = \text{sa}(T, \text{Act})\}$$

be the set of runs such that action $a$ is taken infinitely often in state $s$ iff $s \in T$ and $a \in \text{Act}(s)$. Note that the $\Omega^{(T, \text{Act})}$ constitute a partition of $\text{Runs}(S, A)$.

Therefore, it suffices to establish for any sub-MDP $(T, \text{Act})$, $(T, \text{Act})$ is an end-component or $\mathbb{P}[\rho \in \Omega^{(T, \text{Act})}] = 0$.

Let $(T, \text{Act})$ be an arbitrary sub-MDP. First, suppose there exist $s \in T$ and $a \in \text{Act}(t)$ such that $p := \sum_{s' \in T} \Delta(t, a, t') < 1$. By definition each $\rho \in \Omega^{(T, \text{Act})}$ takes action $a$ in state $s$ infinitely often. Hence, not only $\mathbb{P}[\rho \in \Omega^{(T, \text{Act})}] \leq p^k$ for all $k \in \mathbb{N}$ but also $\mathbb{P}[\rho \in \Omega^{(T, \text{Act})}] = 0$.

Thus, we can assume that for all $s \in T$ and $a \in \text{Act}(t)$, $\sum_{s' \in T} \Delta(t, a, t') = 1$. If $\Omega^{(T, \text{Act})} = \emptyset$ then clearly $\mathbb{P}[\rho \in \Omega^{(T, \text{Act})}] = 0$ follows. Otherwise, take any $\rho = s_0a_0a_1\cdots \in \Omega^{(T, \text{Act})}$, and let $t, t' \in T$ be arbitrary. We show that there exists a connecting path in $(T, \rightarrow_{\text{Act}})$, which implies that $(T, \text{Act})$ is an end component.

Evidently, there exists an index $i_0$ such that all state-action pairs occur infinitely often in $\rho$, i.e.

$$\{(s_{i_0}, a_{i_0}), (s_{i_0+1}, a_{i_0+1}), \ldots\} = \text{InfSA}(\rho)$$

Thus, for all $i \geq i_0$, $s_i \in T$ and $a_i \in \text{Act}(s_i)$, and for all $i' > i \geq i_0$, there is a path from $s_i$ to $s_{i'}$ in $(T, \rightarrow_{\text{Act}})$. Finally, it suffices to note that clearly for some $i' > i = i_0$, $s_i = t$ and $s_{i'} = t'$. □

*Proof of Lemma 8.* By Lemma 21, almost surely $\text{sub}(\text{InfSA}(\rho))$ is an accepting end component. Clearly, $\rho$ is only accepted by the product MDP if this end component is an *accepting* EC. By Lemma 20 this AEC contains an ASEC. Therefore, by definition of $\text{sub}(\text{InfSA}(\rho))$, $\rho$ almost surely in particular *enters* some ASEC. Finally, since the $\mathcal{C}_1, \ldots, \mathcal{C}_n$ cover all states in ASECs, $\rho$ almost surely enters some $\mathcal{C}_i$. □

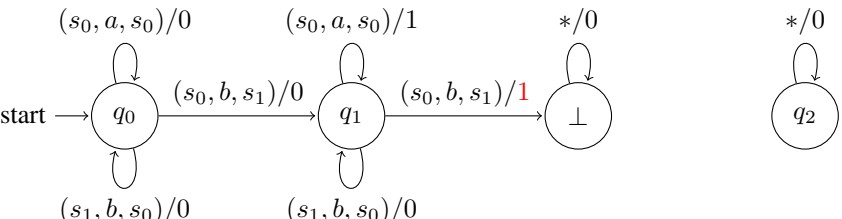

Figure 4: Reward machine yielded by our construction in Section 4 for the running example.

Before turning to the proof of Lemma 10, let $\mathcal{J}^{\mathcal{M}}_{\mathcal{R}^{\text{avg}}}(\rho) = \liminf_{t\to\infty} \frac{1}{t} \cdot \sum_{i=0}^{t-1} r_i$ denote the limit-average reward of a run $\rho$. Note that, for any run $\rho$, $\mathcal{J}^{\mathcal{M}}_{\mathcal{R}^{\text{avg}}}(\rho) \in \{0,1\}$. Thus, by the dominated convergence theorem [28, Cor. 6.26],

$$\mathbb{P}_{\rho\sim\mathcal{D}^{\mathcal{M}}_\pi}\left[\mathcal{J}^{\mathcal{M}}_{\mathcal{R}^{\text{avg}}}(\rho) = 1\right] \;=\; \mathbb{E}_{\rho\sim\mathcal{D}^{\mathcal{M}}_\pi}[\mathcal{J}^{\mathcal{M}}_{\mathcal{R}^{\text{avg}}}(\rho)] \;=\; \liminf_{t\to\infty}\mathbb{E}_{\rho\sim\mathcal{D}^{\mathcal{M}}_\pi}\left[\frac{1}{t}\cdot\sum_{i=0}^{t-1} r_i\right] \;=\; \mathcal{J}^{\mathcal{M}}_{\mathcal{R}^{\text{avg}}}(\pi) \tag{3}$$

**Lemma 10.** *Let $P$ be a probability transition function with support $E$ and $\mathcal{M} := (S, A, s_0, P)$.*

1. *For every policy $\pi$, $\mathcal{J}^{\mathcal{M}}_{\mathcal{R}^{\text{avg}}}(\pi) \leq \mathcal{J}^{\mathcal{M}}_{\mathcal{A}}(\pi)$.*
2. *For every policy $\pi$, there exists some policy $\pi'$ satisfying $\mathcal{J}^{\mathcal{M}}_{\mathcal{A}}(\pi) \leq \mathcal{J}^{\mathcal{M}}_{\mathcal{R}^{\text{avg}}}(\pi')$.*

*Proof.*     1. For any run $\rho$, $\mathcal{J}^{\mathcal{M}}_{\mathcal{R}^{\text{avg}}}(\rho) = 1$ only if $\rho^\otimes$ enters a $\mathcal{C}_i$ and never leaves it. ($\rho^\otimes$ might have entered other $\mathcal{C}_j$'s earlier but then those necessarily need to overlap with yet another $\mathcal{C}_k$ such that $i \leq k < j$ to avoid being trapped in state $\bot$, resulting in $\mathcal{J}^{\mathcal{M}}_{\mathcal{R}^{\text{avg}}}(\rho) = 1$. Furthermore, this $\mathcal{C}_i$ can only overlap with $\mathcal{C}_j$ if $i < j$. Otherwise, the reward machine would have enforced transitioning to $\mathcal{C}_j$.)

Since $\mathcal{C}_i$ is an ASEC, $\rho^\otimes$ is accepted by the product MDP $\mathcal{M} \otimes \mathcal{A}$. Hence, by Eqs. (2) and (3),

$$\mathcal{J}^{\mathcal{M}}_{\mathcal{R}^{\text{avg}}}(\pi) \;=\; \mathbb{P}_{\rho\sim\mathcal{D}^{\mathcal{M}}_\pi}\left[\mathcal{J}^{\mathcal{M}}_{\mathcal{R}^{\text{avg}}}(\rho) = 1\right] \;\leq\; \mathbb{P}_{\rho\sim\mathcal{D}^{\mathcal{M}}_\pi}\left[\rho^\otimes \text{ accepted by } \mathcal{M}\otimes\mathcal{A}\right] \;=\; \mathcal{J}^{\mathcal{M}}_{\mathcal{A}}(\pi)$$

2. Let $\pi$ be arbitrary. For a run $s_0 a_0 \cdots$ let $q_t$ be the state of the DRA in step $t$. Define $\pi'$ to follow $\pi$ until reaching $s_t$ such that $(s_t, q_t) \in T_1 \cup \cdots \cup T_n$. Henceforth, we select the (unique) action guaranteeing to stay in the $\mathcal{C}_i$ with minimal $i$ including the current state, i.e. $\text{Act}_{(q,u)}(q, u)$. Formally[9],

$$\pi'(s_0 a_0 \cdots s_t) \;:=\; \begin{cases} \pi(s_0 a_0 \cdots s_t) & \text{if } (s_t, q_t) \notin T_1 \cup \cdots \cup T_n \\ \text{Act}_{(s_t, q_t)}(s_t, q_t) & \text{otherwise} \end{cases} \tag{4}$$

Note that whenever a run $\rho \sim \mathcal{D}^{\mathcal{M}}_{\pi'}$ follows the modified policy $\pi'$ and its induced run $\rho^\otimes$ reaches some ASEC $\mathcal{C}_i$ then $\mathcal{J}^{\mathcal{M}}_{\mathcal{R}^{\text{avg}}}(\rho) = 1$. Thus,

$$\mathbb{P}_{\rho\sim\mathcal{D}^{\mathcal{M}}_{\pi'}}[\rho^\otimes \text{ reaches some } \mathcal{C}_i] \;\leq\; \mathbb{E}_{\rho\sim\mathcal{D}^{\mathcal{M}}_{\pi'}}[\mathcal{J}^{\mathcal{M}}_{\mathcal{R}^{\text{avg}}}(\rho)] \;=\; \mathcal{J}^{\mathcal{M}}_{\mathcal{R}^{\text{avg}}}(\pi')$$

Furthermore, by Lemma 8 almost surely, every induced run $\rho^\otimes$ accepted by the product MDP must reach some $\mathcal{C}_i$. Consequently, by Eq. (2),

$$\begin{aligned} \mathcal{J}^{\mathcal{M}}_{\mathcal{A}}(\pi) &= \mathbb{P}_{\rho\sim\mathcal{D}^{\mathcal{M}}_\pi}[\rho^\otimes \text{ is accepted by } \mathcal{M}\otimes\mathcal{A}] \\ &\leq \mathbb{P}_{\rho\sim\mathcal{D}^{\mathcal{M}}_\pi}[\rho^\otimes \text{ reaches some } \mathcal{C}_i] \\ &= \mathbb{P}_{\rho\sim\mathcal{D}^{\mathcal{M}}_{\pi'}}[\rho^\otimes \text{ reaches some } \mathcal{C}_i] \leq \mathcal{J}^{\mathcal{M}}_{\mathcal{R}^{\text{avg}}}(\pi') \end{aligned}$$

In the penultimate step, we have exploited the fact that $\pi$ and $\pi'$ agree until reaching the first $\mathcal{C}_i$. $\qquad\square$

---

[9]We slightly abuse notation in the "otherwise"-case and denote by $\text{Act}_{(s_t,q_t)}(s_t, q_t)$ the distribution selecting the state in the singleton set $\text{Act}_{(s_t,q_t)}(s_t, q_t)$ with probability 1.

### B.1 Efficient Construction

We consider a different collection $\mathcal{C}_1, \ldots, \mathcal{C}_n$ of ASECs:

> Suppose $\mathcal{C}'_1, \ldots, \mathcal{C}'_n$ is a collection of AECs (not necessarily simple ones) containing all states in AECs. Then we consider ASECs $\mathcal{C}_1, \ldots, \mathcal{C}_n$ such that $\mathcal{C}_i$ is contained in $\mathcal{C}'_i$.

The definition of the reward machine in Section 4.2 and the extension in Section 5 do not need to be changed. Next, we argue the following:

1. This collection can be obtained efficiently (in time polynomial in the size of the MDP and DRA).

2. Lemma 10 and hence the correctness result (Theorem 9) still hold.

For 1. it is well-known that a collection of maximal AECs (covering all states in AECs) can be found efficiently using graph algorithms [1, Alg. 3.1], [15, 11] and [4, Alg. 47 and Lemma 10.125]. Subsequently, Lemma 20 can be used to obtain an ASEC contained in each of them. In particular, note that the proof of Lemma 20 immediately gives rise to an efficient algorithm. (Briefly, we iteratively remove actions and states whilst querying reachability properties.)

For 2., the first part of Lemma 10 clearly still holds. For the second, we modify policy $\pi$ as follows: Once, $\pi$ enters a maximal accepting end component we select an action on the shortest path to the respective ASEC $\mathcal{C}_i$ inside $\mathcal{C}'_i$. Once we enter one of the $\mathcal{C}_i$ we follow the actions specified by the ASEC as before. Observe that the probability that under an AEC is entered is the same as the probability that one of the $\mathcal{C}_i$ is entered under the modified policy. The lemma, hence Theorem 9, follow.

## C   Supplementary Materials for Section 5

**Lemma 14.** *Suppose $\mathcal{M} = (S, A, s_0, P)$ is an arbitrary MDP.*

1. *For every policy $\pi$, $\mathcal{J}^{\mathcal{M}}_{\mathcal{R}^{avg}}(\pi) \leq \mathcal{J}^{\mathcal{M}}_{\mathcal{A}}(\pi)$.*
2. *For every policy $\pi$, there exists some policy $\pi'$ satisfying $\mathcal{J}^{\mathcal{M}}_{\mathcal{A}}(\pi) \leq \mathcal{J}^{\mathcal{M}}_{\mathcal{R}^{avg}}(\pi')$.*

*Proof.*     1. For a run $\rho$, let $E_\rho$ be the set of transitions encountered in the product MDP. Note that $\mathcal{J}^{\mathcal{M}}_{\mathcal{R}^{avg}}(\rho) = 1$ only if $\rho^\otimes$ enters some $\mathcal{C}_i^{E_\rho}$ and never leaves it. ($\rho^\otimes$ might have entered other $\mathcal{C}_j^E$s earlier for $E \subseteq E_\rho$.)

With probability 1, $E_\rho$ contains all the transitions present in $\mathcal{C}_i^{E_\rho}$ in the actual MDP. (NB possible transitions outside of $\mathcal{C}_i^{E_\rho}$ might be missing from $E_\rho$.) In particular, with probability 1, $\mathcal{C}_i^{E_\rho}$ is also an ASEC for the true unknown MDP and $\rho^\otimes$ is accepted by the product MDP $\mathcal{M} \otimes \mathcal{A}$. Consequently, using Eq. (3) again,

$$\mathcal{J}^{\mathcal{M}}_{\mathcal{R}^{avg}}(\pi) = \mathbb{P}_{\rho \sim \mathcal{D}^{\mathcal{M}}_\pi}[\mathcal{J}^{\mathcal{M}}_{\mathcal{R}^{avg}}(\rho) = 1] \leq \mathbb{P}_{\rho \sim \mathcal{D}^{\mathcal{M}}_\pi}[\rho^\otimes \text{ accepted by } \mathcal{M} \otimes \mathcal{A}] = \mathcal{J}^{\mathcal{M}}_{\mathcal{A}}(\pi)$$

2. Let $\pi$ be arbitrary. We modify $\pi$ to $\pi'$ as follows: until reaching an ASEC $\mathcal{C} = (T, \mathrm{Act})$ w.r.t. the true, unknown[10] set of transitions $E^*$ follow $\pi$. Henceforth, select action $\mathrm{Act}^{E^*}_{(s_t, q_t)}(s_t, q_t)$.

We claim that whenever $\rho \sim \mathcal{D}^{\mathcal{M}}_{\pi'}$ follows the modified policy $\pi'$ and $\rho^\otimes$ reaches some ASEC in the true product MDP, $\mathcal{J}^{\mathcal{M}}_{\mathcal{R}^{avg}}(\rho) = 1$.

To see this, suppose $\rho \sim \mathcal{D}^{\mathcal{M}}_{\pi'}$ is such that for some minimal $t_0 \in \mathbb{N}$, $(s_{t_0}, q_{t_0}) \in T_1^{E^*} \cup \cdots \cup T_n^{E^*}$. Let $\mathcal{C} = (T, \mathrm{Act}) := \mathcal{C}^{E^*}_{(s_{t_0}, q_{t_0})}$.

Define $E_t$ to be the transitions encountered up to step $t \in \mathbb{N}$, i.e. $E_t := \{((s_k, q_k), a_k, (s_{k+1}, q_{k+1})) \mid 0 \leq k < t\}$. Then almost surely for some minimal $t \geq t_0$,

---

[10]NB The modified policy depends on the true, unknown $E^*$; we only claim the *existence* of such a policy.

$E_t$ contains all transitions in $\mathcal{C}$, and no further transitions will be encountered, i.e. for all $t' \geq t$, $E_{t'} = E_t$. Define $\overline{E} := E_t$. Note that for all $((s,q), a, (s', q')) \in \overline{E}$ such that $(s,q) \in T$, $\mathrm{Act}(s,q) = \{a\}$. (This is because upon entering the ASEC $\mathcal{C}$ we immediately switch to following the action dictated by $\mathrm{Act}$. Thus, we avoid "accidentally" discovering other ASECs w.r.t. the partial knowledge of the product MDP's graph, which might otherwise force us to perform actions leaving $\mathcal{C}$.) Consequently, there cannot be another ASEC $\mathcal{C}' = (T', \mathrm{Act}')$ w.r.t. $\overline{E}$ overlapping with $\mathcal{C}$, i.e. $T \cap T' \neq \emptyset$. Therefore, for all $(s,q) \in \mathcal{C}$, $\mathrm{Act}^{\overline{E}}_{(s,q)} = \mathrm{Act}$. Consequently, $\mathcal{J}^{\mathcal{M}}_{\mathcal{R}^{\mathrm{avg}}}(\rho) = 1$.

Thus,

$$\mathbb{P}_{\rho \sim \mathcal{D}^{\mathcal{M}}_{\pi'}}[\rho^{\otimes} \text{ reaches some ASEC in true product MDP}] \leq \mathbb{E}_{\rho \sim \mathcal{D}^{\mathcal{M}}_{\pi'}}[\mathcal{J}^{\mathcal{M}}_{\mathcal{R}^{\mathrm{avg}}}(\rho)] = \mathcal{J}^{\mathcal{M}}_{\mathcal{R}^{\mathrm{avg}}}(\pi')$$

Consequently,

$$\begin{aligned}
\mathcal{J}^{\mathcal{M}}_{\mathcal{A}}(\pi) &= \mathbb{P}_{\rho \sim \mathcal{D}^{\mathcal{M}}_{\pi}}[\rho^{\otimes} \text{ is accepted by } \mathcal{M} \otimes \mathcal{A}] \\
&\leq \mathbb{P}_{\rho \sim \mathcal{D}^{\mathcal{M}}_{\pi}}[\rho^{\otimes} \text{ reaches some ASEC in true product MDP}] \\
&= \mathbb{P}_{\rho \sim \mathcal{D}^{\mathcal{M}}_{\pi'}}[\rho^{\otimes} \text{ reaches some ASEC in true product MDP}] \leq \mathcal{J}^{\mathcal{M}}_{\mathcal{R}^{\mathrm{avg}}}(\pi')
\end{aligned}$$

In the penultimate step we have exploited that $\pi$ and $\pi'$ agree until reaching some ASEC in true product MDP. $\qquad\square$

# D   Supplementary Materials for Section 6

Let $\Pi$ be the set of all memoryless policies and $\Pi^*$ be the set of all limit-average optimal policies. Besides, let $w^* := \mathcal{J}^{\mathcal{M}}_{\mathcal{R}^{\mathrm{avg}}}(\pi^*)$ the limit average reward of any optimal $\pi^* \in \Pi^*$.

Lemma 15 is proven completely analagously to the following (where $K = \mathbb{N}$):

**Lemma 22.** *Suppose $\gamma_k \nearrow 1$, $\epsilon_k \searrow 0$ and each $\pi_k$ is a memoryless policy satisfying $\mathcal{J}^{\mathcal{M}}_{\mathcal{R}^{\gamma_k}}(\pi_k) \geq \mathcal{J}^{\mathcal{M}}_{\mathcal{R}^{\gamma_k}}(\pi) - \epsilon_k$ for all $\pi \in \Pi$. Then there exists $k_0$ such that for all $k \geq k_0$, $\pi_k$ is limit average optimal.*

*Proof.* We define $\Delta := \min_{\pi \in \Pi \setminus \Pi^*} \mathcal{J}^{\mathcal{M}}_{\mathcal{R}^{\mathrm{avg}}}(\pi) - w^* > 0$. Recall (see e.g. [22, Sec. 8.1]) that for any policy $\pi \in \Pi$,

$$\lim_{\gamma \nearrow 1}(1 - \gamma) \cdot \mathcal{J}^{\mathcal{M}}_{\mathcal{R}^{\gamma}}(\pi) = \mathcal{J}^{\mathcal{M}}_{\mathcal{R}^{\mathrm{avg}}}(\pi) \tag{5}$$

Since $\Pi$ is finite, due to Eq. (5) there exists $\gamma_0$ such that

$$|\mathcal{J}^{\mathcal{M}}_{\mathcal{R}^{\mathrm{avg}}}(\pi) - (1 - \gamma) \cdot \mathcal{J}^{\mathcal{M}}_{\mathcal{R}^{\gamma}}(\pi)| \leq \frac{\Delta}{4} \tag{6}$$

for all $\pi \in \Pi$ and $\gamma \in [\gamma_0, 1)$. Let $\pi^*$ be a memoryless Blackwell optimal policy (which exists due to [6, 18]). Note that

$$w^* = \mathcal{J}^{\mathcal{M}}_{\mathcal{R}^{\mathrm{avg}}}(\pi^*) \tag{7}$$

and there exists $\overline{\gamma} \in [0, 1)$ such that

$$\mathcal{J}^{\mathcal{M}}_{\mathcal{R}^{\gamma}}(\pi^*) \geq \mathcal{J}^{\mathcal{M}}_{\mathcal{R}^{\gamma}}(\pi) \tag{8}$$

for all $\gamma \in [\overline{\gamma}, 1)$ and $\pi \in \Pi$. Moreover, there clearly exists $k_0$ such that $\epsilon_k \leq \Delta/4$ and $\gamma_k \geq \gamma_0, \overline{\gamma}$ for all $k \geq k_0$.

Therefore, for any $k \geq k_0$,

$$\begin{aligned}
|\mathcal{J}^{\mathcal{M}}_{\mathcal{R}^{\mathrm{avg}}}(\pi_k) - w^*| &\leq (1 - \gamma_k) \cdot |\mathcal{J}^{\mathcal{M}}_{\mathcal{R}^{\gamma_k}}(\pi_k) - \mathcal{J}^{\mathcal{M}}_{\mathcal{R}^{\gamma_k}}(\pi^*)| + \frac{\Delta}{2} && \text{Eqs. (6) and (7)} \\
&\leq (1 - \gamma_k) \cdot \epsilon_k + \frac{\Delta}{2} && \text{premise and Eq. (8)} \\
&\leq \frac{4}{3} \cdot \Delta
\end{aligned}$$

Consequently, by definition of $\Delta$, $\pi_k \in \Pi^*$. $\qquad\square$

