# OpenReview forum: "Reinforcement Learning with LTL and $\omega$-Regular Objectives via Optimality-Preserving Translation to Average Rewards"
_NeurIPS.cc/2024/Conference — NeurIPS 2024 poster_

### Official Review · Reviewer_EhuA · 2024-07-11

**Soundness:** 4
**Presentation:** 3
**Contribution:** 3
**Rating:** 6
**Confidence:** 4

**Summary:**

The authors present a theoretical framework for learning $\omega$-regular properties in unknown MDPs through average reward optimization (via an "optimality preserving" approach).
Compared to previous work, the approach allows for multi-chain MDP.
The idea is to formalize the property through a deterministic Rabin automaton, construct an auxiliary model (a product MDP) that encodes the state space and the transitions of this automaton into those of the original MDP, and incorporate the resulting model into a reward machine.
Rewards are then generated according to the current knowledge of the underlying graph of the product MDP.
The reward machine can be constructed on the fly, which enables its optimization through RL.
Finally, the authors show that optimizing the discounted return objective boils down to (i) optimizing the average reward objective, which in turn implies (ii) the optimality preserving learning of the $\omega$-regular property.
Finally, convergence guarantees are proved through a sequential algorithm making calls to any PAC-MDP algorithm for discounted return, eventually decreasing the error and confidence terms to 0 as the algorithm progresses.

**Strengths:**

The authors answer questions that were left open: (i) learning $\omega$-regular properties through a limit average reward objective in an optimality-preserving way is not possible with Markovian rewards, (ii) this is in contrast made possible by using finite-spaces reward machines, and (iii) in RL, one can design such an algorithm that converges in the limit.
Overall, the paper is well-written and the proof sketches in the main text effectively allow the reader to grasp the main ideas of the proofs presented in the appendix.

**Weaknesses:**

First, although the paper is quite well written, I think that for somebody unfamiliar with all the concepts presented, the paper might be difficult to follow. For instance, the authors build upon accepting end components to detect when the specification is satisfied. However, the authors do not mention or discuss the reason why it is possible to do that in practice. Specifically, I would expect the authors to mention that identifying ECs can be done through purely graph algorithms, which makes them affordable to consider in this context.

Second, I found that the authors do not really take into account practical considerations in the proposed approach. For instance, there is no mention of how to detect minimal ECs. I mention other issues that I believe should be clarified in the "Questions" section.

Finally, there is no evaluation. I understand that this is intentional due to the theoretical nature of the paper. However, after reading the text, I am eager to know if the approach works in practice, as learning to satisfy LTL specifications is an ongoing challenge in RL. Furthermore, Algorithm 1 is an (unrealizable) theoretical algorithm but I would have loved to see a discussion or remark on how to implement such an algorithm in practice.

**Questions:**

-  Is listing all minimal ECs not equivalent to listing all the possible deterministic policies in a maximal end component? If that's the case, even though the memory of the reward machine need not be too large due to the coverage sufficiency, detecting them might be costly and in the worst case exponential. I believe this question should be addressed and at least discussed in the main text.
- Section 5 is a bit ambiguous to me. In particular, I don't understand how Lemma 13 applies without fixing the set of visited transitions $E$. By looking at the Appendix, I understand that either the set $E$ is fixed, or it changes over time steps, i.e., the set is defined as $E_t$ for $t \geq 0$. If this is the case, then the reward machine $\mathcal{R}$ should take this time step into account in its definition, and I expect to see this in the main text. Could you please clarify?
- Could you please clarify what happens if the current algorithm believes to act in an EC (according to $E$) that is actually _not_ an EC? For instance, what happens if a region of the system is detected at the current time step as being an EC that has a *very* small probability of being left?
- Do you have any insights on how to design a *practical* algorithm that preserves the theoretical guarantees in the limit?

### Remarks and suggestions
- In Section 6, I was confused because I did not understand why considering a discounted return objective is relevant for the average return. I had to check in the appendix to see the reference, mentioning that when the discount $\gamma$ goes to one, $(1 - \gamma) \mathcal{J}\_{\mathcal{R}^{\gamma}}$ coincides with $\mathcal{J}\_{\mathcal{R}^{\text{avg}}}$. This should be explicitly stated in the main text!
- line 131: everyone => every?
- for the reference that LTL is not PAC-learnable, I would also cite [1]
- line 177: it seems that you used $R_i$ for denoting the sets of states visited finitely often and then suddenly switched to $B_i$.

[1] Hugo Bazille, Blaise Genest, Cyrille Jégourel, Jun Sun: Global PAC Bounds for Learning Discrete Time Markov Chains. CAV (2) 2020: 304-326

**Limitations:**

Limitations have been addressed and further discussed in the conclusion.

---

> ### Author Rebuttal · Authors · 2024-08-07
>
> We thank the reviewer for their insightful comments. We will take their suggestions into account when revising the paper. In particular, we will use the additional page for the camera-ready version to elaborate on practical aspects and the construction in Sec. 5.
> Next, we address the reviewer’s questions (numbered 1-4).
>
> # Practical Aspects: Constructing the Reward Machine (Question 1)
> ASECs yield (partial) deterministic policies. Our construction considers a collection of ASECs covering all states in ASECs (l. 205). It does not necessarily require listing all possible ASECs but only (up to) one ASEC per state. It is unclear whether this can be obtained in polynomial time but the focus of our submission was primarily simplicity of presentation and clarity of the optimality-preserving translation. In the following, we present a slightly modified construction which is *efficient*. We will give a brief account in the revision.
>
> We consider a different collection $\mathcal C_1,\ldots,C_n$ of ASECs:
> > Suppose $\mathcal C’_1,\ldots,\mathcal C’_n$ is a collection of AECs (not necessarily simple ones) containing all states in AECs. Then we consider ASECs $\mathcal C_1,\ldots,C_n$ such that $\mathcal C_i$ is contained in $\mathcal C’_i$.
>
> The definition of the reward machine in Sec. 4.2 and the extension in Sec. 5 do not need to be changed. Next, we argue the following:
> 1. This collection can be obtained efficiently (in time polynomial in the size of the MDP and DRA).
> 2. Lemma 10 and hence the correctness result (Thm. 10) still hold.
>
> For 1. it is well-known that a collection $\mathcal C’_1,\ldots,\mathcal C’_n$ of maximal AECs (covering all states in AECs) can be found efficiently using graph algorithms [1, Alg. 3.1], [2,3] and [4, Alg. 47 and Lemma 10.125].
> Subsequently, Lemma 19 can be used to obtain an ASEC contained in each of them. In particular, note that the proof of Lemma 19 immediately gives rise to an efficient algorithm. (Briefly, we iteratively remove actions and states whilst querying reachability properties.)
>
> For 2., the first part of Lemma 10 clearly still holds. For the second, we modify policy $\pi$ as follows: Once, $\pi$ enters a maximal accepting end component we select an action on the shortest path to the respective ASEC $\mathcal C_i$ inside $\mathcal C_i’$. Once we enter one of the $\mathcal C_i$ we follow the actions specified by the ASEC as before. Observe that the probability that under $\pi$ an AEC is entered is the same as the probability that one of the $\mathcal C_i$ is entered under the modified policy. The lemma, hence Thm. 10, follow.
>
> [1] Luca Alfaro. Formal Verification of Probabilistic Systems. PhD thesis, 1998
>
> [2] Fu & Topcu. Probably approximately correct MDP learning and control with Temporal Logic constraints. 2014
>
> [3] Chatterjee & Henzinger, “Faster and dynamic algorithms for maximal end-component decomposition and related graph problems in probabilistic verification”, 2011
>
> [4] Baier & Katoen. Principles of Model Checking. 2008
>
> # Construction in Sec. 5 (Question 2/3)
> Intuitively, in Sec. 5 we can eliminate knowledge of the transitions with positive probability because for a run we almost surely see enough transitions of the true product MDP to determine whether it reaches and stays in an AEC.
> This is due to the well-known result that with probability 1, the states and actions occurring infinitely often in runs constitute an end component (not necessarily an accepting one) w.r.t. the true MDP (Lemma 20). Moreover, for accepting runs this EC must clearly be accepting.
>
> Therefore, our reward machine tracks the set $E$ of transitions in the product MDP encountered earlier in a run as part of its state, and $E$ is used to assign rewards, which enforce staying in one of the dedicated ASEC relative to the current knowledge $E$. Whenever a new transition is seen, the RM resets the status flag and we will need to follow one of the ASECs w.r.t. the updated $E$. Importantly, in the setting of limit-average rewards, initial “missteps” do not affect the total reward.
>
> ## Question 2
> Further to the above explanations, we illustrate the evolvement of the $E$-component of the RM’s state over time:
> Initially, this $E$-component of the state is $\emptyset$. Over time, as new transitions are observed in a run, $E$ grows monotonically. Since the MDP is finite, at some time step all transitions in a run have been seen and $E$ does not grow further.
>
> The reward machine in Sec. 5 does take the current $E$ into account and rewards transitions based on the dedicated ASECs w.r.t. $E$. (Note the superscripts in the definition following l. 257.) There is no need to consider the number $t$ of the time step.
>
> ## Question 3
> Suppose we follow one of the dedicated ASECs w.r.t. the transitions observed in the past (and hence receive rewards of 1). If this is not ASEC w.r.t. the true product MDP there exists a transition leaving it. As argued above, almost surely it will eventually be taken. (This may take a while if it has very small probability.)
> Thereafter, the reward machine acts according to the updated set of possible transitions. The rewards obtained thus far do not affect the overall limit-average reward.
>
> # Practical Algorithm with Convergence Guarantee (Question 4)
> Algorithm 1 can be improved in practice by exploiting the internals of the RL-algorithm ```Discounted``` with PAC-guarantees and running it incrementally. For example, the model-based approaches of [5,6] first sample a sufficiently good approximation of the MDP’s transition probabilities before planning optimal policies on it. Instead of discarding earlier samples on the next invocation of ```Discounted```, we can simply determine how many additional samples are required to achieve the more stringent guarantees for the updated parameters.
>
> [5] Kearns & Singh. Near-optimal reinforcement learning in polynomial time. 2002
>
> [6] Strehl et al. Reinforcement learning in finite MDPs: PAC analysis. 2009

---

> > ### Comment · Reviewer_EhuA · 2024-08-12
> >
> > Thank you for the clarification. However, I still have a question. You mention that with probability one, if $E$ is not an EC, the algorithm should see the leaving transition. This is true, however with a PAC algorithm, you are given a finite number of samples. What happens if the transition is never seen? Can you bound the maximum error that this may incur?
> >
> > Concerning the practical algorithm, this is interesting. I still have a question though; if you use the knowledge from the previous iterations, then each call to the PAC algorithm is dependent on the previous one. Is that not a problem?

---

> ### Author Response · Authors · 2024-08-12
>
> We thank the reviewer for their ongoing discussion.
> > However, I still have a question. You mention that with probability one, if $E$ is not an EC, the algorithm should see the leaving transition. This is true, however with a PAC algorithm, you are given a finite number of samples. What happens if the transition is never seen? Can you bound the maximum error that this may incur?
>
> If we understand this question correctly, it is concerned about the correctness of the reward machine in Sec. 5, i.e. that the translation is optimality-preserving, as formalised in Thm. 9 and Lemma 10.
> We believe there might be some conceptual misunderstanding: in the present paper we do not discuss algorithms that directly solve the RL problem with $\omega$-regular/LTL-objectives. Rather, we present an *optimality-preserving translation* to the more standard problem of RL with limit-average reward machines. Furthermore, we underpin the efficacy of this approach by providing an algorithm with theoretical guarantees for the latter problem. Crucially, the algorithm for limit-average reward machines is independent of the translation from $\omega$-regular/LTL-rewards.
>
> We stress that for the *construction* of the reward machine we do not sample from the MDP. On the other hand, for the correctness of the reward machine we need to ensure that almost all *runs* are accepted by the product MDP if they receive a limit-average reward of 1 (Lem. 10.1).
> It is important to note that since runs have infinite length they contain an infinite number of transitions. Hence, the probability that a run never takes a transition leaving a suspected EC is $0$.
>
>
>
> > Concerning the practical algorithm, this is interesting. I still have a question though; if you use the knowledge from the previous iterations, then each call to the PAC algorithm is dependent on the previous one. Is that not a problem?
>
> Indeed, if the calls in Alg. 1 are incremental (re-using samples of earlier iterations), the failure probabilities in the iterations are no longer independent. However, for correctness of the overall algorithm we only need that the expected number of failures is finite. This is proven in Thm. 15 exploiting linearity of expectations, which even holds for non-independent random variables.

---

### Official Review · Reviewer_eqsH · 2024-07-12

**Soundness:** 4
**Presentation:** 4
**Contribution:** 4
**Rating:** 7
**Confidence:** 4

**Summary:**

The paper studies the link between reinforcement learning with $\omega$-regular objectives to reinforcement learning with limit-average reward. It is shown that one cannot reduce RL with $omega$-regular objectives to RL with limit-average objectives by only replacing the reward function (Proposition 4), but that it is possible if one considers rewards given by finite-memory reward machines (Theorem 11), which is the main result of the paper. It is also shown that RL with limit-average rewards can be learned in the limit, thus implying that RL with omega-regular objectives can also be learned in the limit (Theorem 17).

**Strengths:**

The paper is original in the sense that it provides theoretical tools for a new approach for RL with $\omega$-regular objectives by reducing it to the well-known case of limit-average rewards. This allows to apply the many approaches designed originally for the case of limit-average rewards to $\omega$-regular objectives. The idea of approaching RL with $\omega$-regular objectives by analyzing the end components of the product MDP with a DRA or LDBA representing the objective might not be new, but the way it is precisely used with accepting simple components and reward machines is new and original to the best of my knowledge. I would judge the paper of high quality as it is well presented, the proofs are sound as far as I could check, and the context of the results is very well explained. The paper is easy to read and its more technical concepts are explained in a simple manner (Section 4 is a good addition in that sense). The results of the paper solves open problems given in [1], and contribute to building a theoretical framework for transforming specifications in RL, which is right now underdeveloped with many open questions, and was a work begun in [1]. Moreover, even though no practical applications of the results are presented, it is not unreasonable to expect that the main result might be directly applied in the case of finite MDPs with the support of the actions known. Therefore, the results are of high significance to the community.

[1]: Rajeev Alur, Suguman Bansal, Osbert Bastani, and Kishor Jothimurugan. A framework for transforming specifications in reinforcement learning. In Jean-François Raskin, Krishnendu Chatterjee, Laurent Doyen, and Rupak Majumdar, editors, Principles of Systems Design: Essays Dedicated to Thomas A. Henzinger on the Occasion of His 60th Birthday, pages 604–624, Cham, 2022. Springer Nature Switzerland.

**Weaknesses:**

I do not think that the paper has any obvious weaknesses, but it could be improved by either providing a polynomial construction for the reduction from $\omega$-regular objectives to limit-average ones in the case where the probabilities of the MDP are unknown or providing a negative result.

**Questions:**

I do not have any questions

**Limitations:**

The authors addressed the limitations of their work in the Limitations section at the end.

---

> ### Author Rebuttal · Authors · 2024-08-06
>
> We thank the reviewer for their insightful comments.
>
> We agree with the reviewer that investigating the possibility of a polynomial translation for the general case (where the transitions with positive probability are not known) is a very interesting future direction.

---

> > ### Comment · Reviewer_eqsH · 2024-08-12
> >
> > I'm happy with the answers provided by the authors.

---

### Official Review · Reviewer_FqP1 · 2024-07-13

**Soundness:** 4
**Presentation:** 3
**Contribution:** 3
**Rating:** 7
**Confidence:** 3

**Summary:**

This paper tackle several open problems in the field of specification driven learning using Reinforcement Learning (RL) algorithms. It proves that $\omega$-regular objectives can be translated in an optimality preserving manner to limit-average reward MDPs. A PAC-MDP convergence proof for limit-average reward MDPs is introduced as well.

**Strengths:**

The paper reads well and presents a clear and logical progression towards the main results.

**Weaknesses:**

The claim of the first proof of convergence for average reward MDPs may need to be further evaluated [1].

**Questions:**

1. How does the choice between Limit Deterministic Büchi Automatas (LDBAs) and DRAs to express $\omega$-regular languages affect the results [2]?
2. Although [1] does not consider the PAC-MDP setting, how relevant is their proof of convergence w.r.t. the results in the paper?
3. Minor typos
    1.  L131 everyone → every
    2. L156 $\delta_d$ ->  $\delta_u$ ?
    3. L289 “sequence of discount sum”

References:

[1] Learning and Planning in Average-Reward Markov Decision Processes, Wan et al, ICML 2021

[2] From LTL to your favourite deterministic automaton, Kretínsky et al,
In CAV (1), volume 10981 of Lecture Notes in Computer
Science, pages 567–577. Springer, 2018.

**Limitations:**

The cost of running these transformations is not discussed (empirically or otherwise).

---

> ### Author Rebuttal · Authors · 2024-08-07
>
> We thank the reviewer for their insightful comments and we will take their suggestions into account when revising the paper. We proceed to address the questions raised by the reviewer.
>
>
> # Choice of DRAs vs. LDBAs
>
> > How does the choice between Limit Deterministic Büchi Automata (LDBAs) and DRAs to express \Omega-regular languages affect the results?
>
> Our construction of the reward machine relies on the representation of $\omega$-regular languages by DRAs (particularly their determinism). We do not know if there is an *efficient* reduction from LDBAs to limit-average reward machines. Note the presence of non-deterministic transitions in LDBAs, whereas reward machines are inherently deterministic. We will investigate if an efficient reduction from LDBAs to limit-average reward machines is possible avoiding the blow-up incurred from translating LDBAs to DRAs.
>
>
>
> # Novelty of convergence for limit-average objectives
>
> > Although [1] does not consider the PAC-MDP setting, how relevant is their proof of convergence w.r.t. the results in the paper?
>
> Wan et al. [1] require the following (communicating) assumption: “For every pair of states, there exists a policy that transitions from one to the other in a finite number of steps with non-zero probability.”
>
> This assumption generally fails for our setting, where in view of our Cor. 16, MDP states also track the states of the reward machine. For instance, in the reward machine in Fig. 2(a), it is impossible to reach $u_1$ from $u_2$.
>
> Therefore, the paper [1] does not undermine the novelty of our approach for general MDPs.
>
> We will add the reference to the list of similar works discussed in l. 341.
>
> [1] Wan et al.: Learning and Planning in Average-Reward Markov Decision Processes, ICML 2021
>
> [2] Kretínsky et al: Rabinizer 4: From LTL to your favourite deterministic automaton, CAV 2018.

---

> > ### Comment · Reviewer_FqP1 · 2024-08-12
> >
> > I thank the authors for their response and have no further questions.

---

### Official Review · Reviewer_VjJR · 2024-07-13

**Soundness:** 3
**Presentation:** 3
**Contribution:** 2
**Rating:** 6
**Confidence:** 3

**Summary:**

This paper considers the problem of reducing temporal logic objectives to limit-average reward problems which enable using reinforcement learning to compute policies. The key contributions are an explicit construction and analysis for the setting where the MDPs transition support is known, followed by a relaxation which drops this assumption. The key theorems then provide a characterization of the optimality preserving nature of the reduction.

**Strengths:**

The constructions and theorems are to my knowledge novel and sound. Further, the target problem of LTL policy synthesis in the model free setting is important and relevant to the Neurips audience.

Particularly interesting is the adaptive relaxation of the transition support knowledge. Further lemma 13's robustness implications further speak to the potential.

**Weaknesses:**

My primaries concerns for this paper are the (lack of) relation to prior work and the lack of empirical validation, even on a toy problem.

In particular, this works goal and conclusions seem very similar to [1] which also shows a non-markovian reduction policy learning which seems to also be optimal policy preserving with high probability and seems to apply to a larger problem space.

[1] Eventual Discounting Temporal Logic Counterfactual Experience Replay, 2023

If this relationship can be clarified, I could raise my rating.

My other nitpick is with the novelty of section 3. My understanding is that this result is known, for example [2].

[2] On the Expressivity of Markov Reward, 2022

**Questions:**

See weaknesses.

**Limitations:**

Limitations section in paper is satisfactory.

---

> ### Author Rebuttal · Authors · 2024-08-06
>
> We thank the reviewer for their insightful comments. We will take their suggestions into account when revising the paper. In particular, we will clarify the relation to the additional references as detailed below.
>
> # Translation to Eventual Discounted Rewards [1]
>
> > In particular, this work's goal and conclusions seem very similar to [1] which also shows a non-markovian reduction policy learning which seems to also be optimal policy preserving with high probability and seems to apply to a larger problem space.
>
> Voloshin et al. [1] present a translation from LTL objectives to eventual discounted rewards, where only strictly positive rewards are discounted. (In particular, they do not consider average rewards.)
> Their main result (Thm. 4.2) is that
> $$p-p’\leq 2\log(1/\gamma)\cdot m &emsp;&emsp; (*)$$
> where
> - $\gamma$ is the discount factor,
> - $p$ is the optimal probability of LTL satisfaction,
> - $p’$ is the probability of LTL satisfaction by the policy maximising eventual $\gamma$-discounted rewards, and
> - $m:=\sup_\pi O_\pi$ is a constant which depends on the probabilities in the MDP.
>
> This provides a bound on the *sub-optimality* w.r.t. LTL satisfaction of optimal policies for the eventual discounted problem for a given discount factor.
>
> Besides, Cor. 4.5 (of [1]) concludes that for a given error-tolerance $\epsilon$ a discount factor can be chosen guaranteeing a sub-optimality bound (*) above of $\epsilon$, ***provided $m$ is known***. However, a priori, the constant $m$ is not available without knowledge of the MDP.
> As such, this paper does not provide an optimality preserving translation which works regardless of unknown parameters of the MDP. Thus, it falls in a similar category as the approaches mentioned in ll. 335-9 in our discussion of related work.
>
> # Expressivity of Markov Reward
>
> > My other nitpick is with the novelty of section 3. My understanding is that this result is known, for example [2,3]
>
> [2,3] study MDPs with discounted rewards and three types of task specifications: (i) a set of acceptable policies, (ii) a partial ordering on policies, and (iii) a partial ordering on runs.
> Whilst their paper is concerned with the limited expressivity of reward functions, it neither covers LTL nor average rewards.
> In particular, we do not think that our Proposition 4 is stated (or follows from the results) in [2,3]. They acknowledge this by stating the following when briefly mentioning LTL task specifications: “​​A natural direction for future work broadens our analysis to include these kinds of tasks.”
>
> (Task specification (iii)—imposing a partial ordering on runs—is most closely related to LTL tasks. However, their framework enables the expression of far richer preferences. In particular, the task designer can distinguish all traces and is not constrained by the limited expressivity of LTL. This naturally makes finding a suitable reward function even significantly harder.)
>
> [1] Cameron Voloshin, Abhinav Verma, Yisong Yue: Eventual Discounting Temporal Logic Counterfactual Experience Replay. ICML 2023.
>
> [2] David Abel, Will Dabney, Anna Harutyunyan, Mark K. Ho, Michael L. Littman, Doina Precup, Satinder Singh: On the Expressivity of Markov Reward (Extended Abstract). IJCAI 2022
>
> [3] David Abel, Will Dabney, Anna Harutyunyan, Mark K. Ho, Michael L. Littman, Doina Precup, Satinder Singh: On the Expressivity of Markov Reward. NeurIPS 2021

---

> > ### Comment · Reviewer_VjJR · 2024-08-13
> >
> > Thank you for the clarifications.
> >
> > After reading the rebuttal and the other reviews, I’m inclined to revise my score to be higher.
> >
> > I would encourage the authors to add the comparison and contextualization given in their rebuttal to the final draft if possible.

---

> ### Author Response · Authors · 2024-08-13
>
> Dear Reviewer VjJR,
>
> we thank the reviewer for their discussion. We will follow their recommendation and add the comparison and contextualisation when preparing the final revision.
>
> We also appreciate the reviewer's indication to raise their score.

---

### Decision · Program_Chairs · 2024-09-25

**Decision:**

Accept (poster)

**Comment:**

There is common agreement that the paper is making a novel and solid theoretical contribution to the RL area. The reviewers raised several issues that the authors should work to clarify in the final paper.